# RANTES and CD40L under Conditions of Long-Term Physical Exercise: A Potential Link to Adaptive Immunity

**DOI:** 10.3390/ijerph19148658

**Published:** 2022-07-16

**Authors:** Max Lenz, Robert Schönbauer, Stefan Stojkovic, Jonghui Lee, Constantin Gatterer, Michael Lichtenauer, Vera Paar, Michael Emich, Monika Fritzer-Szekeres, Jeanette Strametz-Juranek, Senta Graf, Michael Sponder

**Affiliations:** 1Division of Cardiology, Department of Internal Medicine II, Medical University of Vienna, 1090 Vienna, Austria; max.lenz@meduniwien.ac.at (M.L.); robert.schoenbauer@meduniwien.ac.at (R.S.); stefan.stojkovic@meduniwien.ac.at (S.S.); jonghui.lee@meduniwien.ac.at (J.L.); constantin.gatterer@meduniwien.ac.at (C.G.); senta.graf@meduniwien.ac.at (S.G.); 2Ludwig Boltzmann Cluster for Cardiovascular Research, 1090 Vienna, Austria; 3Clinic of Internal Medicine II, Department of Cardiology, Paracelsus Medical University of Salzburg, 5020 Salzburg, Austria; m.lichtenauer@salk.at (M.L.); v.paar@salk.at (V.P.); 4Austrian Federal Ministry of Defence, Austrian Armed Forces, 1090 Vienna, Austria; dr@emich.at; 5Chemical Laboratory Analysis, Department of Medical, Medical University of Vienna, 1090 Vienna, Austria; monika.fritzer-szekeres@meduniwien.ac.at; 6Rehabilitation Centre Bad Tatzmannsdorf, 7431 Bad Tatzmannsdorf, Austria; dr.jstrametz@sternvilla.com

**Keywords:** sport, training, CCL5, sCD40L, inflammation, immune system, 2022

## Abstract

Regular physical exercise was found to be associated with an improved immune response in previous studies. RANTES and CD40L play a pivotal role in host defense, and individuals lacking adequate expression are prone to virus and opportunistic infections. A total of 98 participants were enrolled in this study. The probands were asked to perform moderate physical activity, and bicycle stress tests were performed at the baseline and after 8 months of training to evaluate individual performance. RANTES and CD40L were found to be increased by long-term physical exercise. In particular, probands with a performance gain of ≥3% displayed a pronounced elevation of both markers, paired with a decrease in circulating IL6 levels and an improved lipid profile. In summary, we were able to highlight rising levels of serum RANTES and CD40L under the conditions of physical exercise. Taking their role in host defense into account, a conjunction of physical activity and the adaptive immune system could therefore be assumed. Furthermore, low inflammatory profiles in probands with a significant performance gain suggest a modulation through exercise rather than a generalized pro-inflammatory status.

## 1. Introduction

The link between physical activity and functions of the immune system has been a topic for more than a century. Early stages in the field focused on changes in immune cell counts and functional assays. With the emergence of flow cytometry in the 1970s, more specific cell characterization was possible. Later, nutrition, inflammation, and various cytokines would attract the scientific community’s attention. The recent past was shaped by multi-omics analyses, personalized medicine, and the understanding of inflammatory ramifications [1]. In light of the current COVID-19 pandemic, the implications of physical activity on host defense seem more crucial than ever [2]. Therefore, a deeper understanding of biomarkers linking immune response to physical activity appears desirable.

RANTES (regulated on activation, normal T-cell expressed and secreted), also known as CCL5 (chemokine C-C motif ligand 5), is a chemokine encoded by the CCL5 gene located on chromosome 17 within the human genome [3]. RANTES exerts an active role in recruiting leukocytes to the site of inflammation as well as modulating the occurring immune response [4]. In addition to the involvement in atherosclerotic plaque formation and local inflammatory processes, it was found to play a role in host defense and regulating the specific immune system [5,6,7]. The work of Cully et al. emphasized the importance of RANTES in viral respiratory infections as competitive blockage led to a hampered inflammatory reaction and a delayed viral clearance [8]. Additionally, in macrophages, evidence for antiapoptotic signaling of RANTES–CCR5 interaction was provided for the setting of acute infection [9]. Ank et al. showed an age-dependent role of RANTES/CCR5 in host defense against HSV-2 (herpes simplex virus 2) by supporting both the specific and nonspecific immune responses [10]. A study conducted in 2014 revealed a decreased expression of RANTES and CCR5 in adipose tissue of obese humans after 3 months of supervised physical exercise [11]. Hoff et al. described an increase in RANTES under the conditions of bed rest (simulated weightlessness) and exercise for 60 days in 17 male probands [12].

CD40L is encoded by the CD40L gene expressed on human chromosome Xq24. It is transiently expressed on activated T-cells and other non-immune cells under inflammatory conditions [13,14]. In addition to increasing the expression of adhesion molecules, macrophage activation via CD40L/CD40 was found to result in an improved phagocytic capacity and cytokine output [15,16]. Wang et al. depicted CD40L as a potential biomarker for atherosclerotic instability [17]. However, its significance in mounting a specific immune response appears well documented and of high importance. Hyper IgM syndrome is caused by a lack of functional CD40L, resulting in a severely impaired B-cell response and opportunistic infections [18]. Further studies emphasized the role in various infectious conditions such as tuberculosis, protozoan, and viral infections [19,20,21]. Short-term ultra-endurance exercise in athletes decreased CD40L for up to 48 h [22]. Bjørnstad et al. reported a decrease in 15 patients with chronic heart failure after a 20-week training regime [23].

The influence of increased physical activity on inflammation and lipid metabolism biomarkers such as sTWEAK/sCD163, sRAGE, and adipsin/ANGPTL4 has already been investigated by our group [24,25,26]. In the present study, we aimed to illuminate the effect of long-term exercise on RANTES and CD40L. Both parameters are linked to inflammatory processes, particularly by emphasizing specific immune responses. Both markers might be viewed as a double-edged sword as they are necessary for a competent host defense but are overexpressed in several inflammatory conditions. Although there is the mentioned evidence on the short-term effects of physical exercise in smaller collectives, more longitudinal data seem to be lacking. In contrast to the work of Hoff et al. as well as Bjørnstad et al., the present study provides a significantly longer observational period in a cohort of probands displaying a heterogeneous risk profile [12,23]. We evaluated the changes of RANTES and CD40L in 98 participants undergoing 8 months of physical exercise. To our knowledge, we thereby provide the longest observational window for both biomarkers in the largest present cohort, so far. Moreover, bicycle stress tests were performed at the baseline and end of the study to set the actual performance levels in perspective to the circulating concentrations of both parameters. Potential findings of the present study could shed some light on the link of long-term physical exercise and the adaptive immune system. Especially enhanced levels of RANTES and CD40L might hint towards a beneficial influence on exerted immune responses such as viral clearance and increased resilience to tuberculosis as well as other infections.

## 2. Materials and Methods

### 2.1. Characterization of the Study Cohort

A total of 109 probands took part in the presented study. Voluntary participants were recruited via pinboard notices providing information of study design and inclusion criteria. No monetary incentives were offered for participation in this study. Inclusion criteria were defined as the mental and physical capability to undergo bicycle stress tests and continuous endurance training for the duration of the study period. Additionally, at least one established cardiovascular risk factor was required for inclusion. Risk factors were defined as: Chronic heart disease (CHD) with prior myocardial infarction, CABG, PCI, or stroke.Family history of cardiovascular disease or stroke in first-degree relatives (only mother or father).The presence of one or more metabolic risk factors including overweight (BMI > 25), diabetes mellitus (HbA1c > 6.5% or the presence of antidiabetic medication), dyslipidemia (marked by statin intake), and arterial hypertension (resting SBP > 140 mmHg/resting DBP > 85 mmHg or the presence of antihypertensive medication).A positive smoking status.

Besides the mentioned inclusion criteria, the presence of an infectious or oncological disease was defined as an exclusion criterion. To ensure the physical requirements for participation, all probands underwent a physical examination at the baseline. Medical and anthropometric data were recorded during the process, and a diagnostic scale (Beurer BG 16, Beurer GmbH, Ulm, Germany) was used for the measurement of the individual body composition. Although all participants were asked to maintain their eating habits unaltered, thorough control of this factor was not possible. Throughout the study period, 11 probands dropped out (10.1%), resulting in the final analyses being conducted on the remaining 98 individuals. 

### 2.2. Performance and Continuous Physical Activity

Individual performance levels at the baseline were determined by a bicycle stress test. For the subsequent study period of 8 months, all probands were instructed to perform moderate physical activity for at least 150 min per week (defined as an intensity level of 65–75% of the maximum heart rate, calculated by Karvonen’s formula) and/or vigorous-intensity exercise for at least 75 min per week (defined as an intensity level of 76–93%). Participants were free to choose the type of exercise, with strength training not being mandatory. Regular follow-ups were scheduled after 2, 6, and 8 months and included acquiring blood samples for routine laboratory measurements and determining circulating RANTES and CD40L levels. A second bicycle stress test was conducted to evaluate changes in individual performance at the end of the study.

### 2.3. Bicycle Stress Tests

All individual tests were performed on the same system (Ergometer eBike comfort, GE Medical Systems, Freiburg, Germany) to reduce potential bias caused by varying investigation modalities. A prior established protocol was used: initial resistance of 25 watts, followed by an increase of 25 watts, added every 2 min. The protocol was designed in accordance with the protocol of the Austrian Society of Cardiology and the guidelines of the European Society of Cardiology. Appropriate monitoring was present throughout the whole test and consisted of an ECG and blood pressure measurements every 2 min. Probands were instructed to cycle with 50–70 revolutions per minute until the occurrence of exhaustion. Sex, age, and body surface (calculated according to the DuBois formula) were utilized to calculate individual target performance with an output of 100%, representing the performance of an untrained collective [27]. Probands were asked to keep a training diary, which was evaluated during each study visit to check the amount of conducted physical exercise. However, since the physical exercise was exerted without supervision, a thorough control of adherence was not possible. Therefore, a second bicycle stress test was performed at the end of the study to quantify changes in performance gain objectively. 

### 2.4. Formation of Groups and Subgroups in Accordance with Individual Performance

Probands were stratified into two major groups as well as four subgroups. A performance level of 100% was assumed for an untrained collective. Participants with an initial performance below 100% were labeled initially unathletic, whereas individuals above 100% were referred to as initially athletic. By utilizing a second bicycle stress test at the end of the study, performance gain was deduced. Furthermore, a performance gain of 3% was deemed a significant improvement. This threshold was selected for two reasons. First, changes in body composition and laboratory values were observable when the 3% threshold was breached, as individuals displayed changes classically associated with continuous physical activity. In this regard, probands above 3% showed decreased body fat and increased HDL levels. Second, this cutoff resulted in a balanced average performance gain of 12% regardless of the initial performance. An overview of the depicted groups can be found below:Group A = subgroups α and γ (*n* = 27)Group B = subgroups β and δ (*n* = 71)
○α: initially unathletic (initial performance < 100%), performance gain ≤ 2.9% (*n* = 9).○β: initially unathletic (initial performance < 100%), performance gain > 2.9% (*n* = 32).○γ: initially athletic (initial performance ≥ 100%), performance gain ≤ 2.9% (*n* = 18).○δ: initially athletic (initial performance ≥ 100%), performance gain > 2.9% (*n* = 39).


The applied method of stratification allowed comparison among the groups and subgroups. Group B (subgroups β and δ) was defined as the “responder” group, in which physical exercise led to anthropometric changes typically associated with training and increased overall performance of ≥3%. Group A (subgroups α and γ) served as a “non-responder” group since no significant performance gain (<3%) or accompanying changes in body composition could be detected.

### 2.5. Laboratory Analysis, RANTES, and CD40L ELISA

Venous blood was drawn at the baseline and during each routinely scheduled visit (after 2, 6, and 8 months). Standard laboratory measurements were performed in addition to determining RANTES and CD40L values via ELISA (DY278, DY617, R&D Systems, Minneapolis, Minn., USA) following the manufacturer’s instructions. Measurements of optical density (OD) were taken on a suitable plate reader (iMark Microplate Absorbance Reader, Bio-Rad Laboratories, Vienna, Austria) at a wavelength of 450 nm. The coefficient of variation (CV) was calculated for RANTES (2.1–9.4% intra-assay and 1.1–9.4% inter-assay) and CD40L (1.9–7.1% intra-assay and 4.1–7.9% inter-assay).

### 2.6. Statistical Analysis

SPSS Statistics 26.0 (IBM, USA) was used for all statistical analyses. Continuous variables are presented as mean ± standard deviation (STDV). Non-normally distributed variables (determined by the Kolmogorov–Smirnov test) are given as median and interquartile ranges (25th–75th quartile). Spearman’s correlation was used to identify correlations between RANTES or CD40L and anthropometric data/laboratory parameters. Found correlations were subsequently included in a linear regression model (input method). The Mann–Whitney U test was used to test for differences in circulating levels of RANTES and CD40L in the presence of different cardiovascular risk factors. Furthermore, the Friedman test was utilized to evaluate the progression of the parameters over the study period. To test for significant differences between the baseline and end of the study, a Wilcoxon signed-rank test was performed. Multiple group comparison was achieved by using one-way ANOVA with Bonferroni correction. Statistical analyses were exclusively performed on non-standardized data for the purpose of better representability; in figures with multiple parameters, data are presented as percentages of the baseline values. The *p*-values of *p* < 0.05 (two-tailed) were considered statistically significant unless indicated otherwise.

## 3. Results

### 3.1. Baseline Characteristics and Performance Gain

Subgroup characteristics, body composition data, cardiovascular risk factors, and laboratory values were recorded at the baseline and are displayed in Table 1. Predominant risk factors were identified in the presence of overweight, hypertension, dyslipidemia, and diabetes mellitus. Within the total study cohort, the mean baseline performance was determined to be 105.6 ± 19.7% (175.0 ± 48.6 watts). Comparing these findings to the measurements at the end of the study (113.7 ± 20.0%, 187.6 ± 53.0 watts), an average increase of 7.8 ± 9.1% (12.6 ± 14.7 watts) was observed. As previously described, subgroups were formed by comparing initial performance and performance gain over the study period. Performance stratification within the individual groups is depicted in Table 1. No significant performance gain was detectable in subgroups α and γ, whereas in subgroups β and δ, an improvement of 12.2 ± 7.1% (19.7 ± 11.5 watts) and 12.1 ± 5.6% (19.5 ± 9.0 watts) was noted at the end of the study. 

### 3.2. RANTES and CD40L throughout the Study

Comparing RANTES values in the total population at the baseline (2.08 ± 2.14 ng/mL, displayed as 100% in Figure 1A) with the final measurements (3.13 ± 2.42 ng/mL, displayed as percentage of the baseline in Figure 1A), an average gain of 50.48% was observed (*p* = 0.02, Figure 1A). Although three out of four subgroups displayed markedly enhanced levels at the end of the study, these results were mostly driven by the pronounced increase in subgroup δ (1.88 ± 1.37 ng/mL vs. 4.04 ± 3.13 ng/mL displayed as percentage of the baseline, *p* < 0.001, Figure 2A). When stratifying for participants with a significant performance gain of ≥3% throughout the study (group B = subgroups β and δ), the picture seems to become clearer as described individuals showed clearly elevated RANTES values (2.00 ± 1.48 ng/mL vs. 3.43 ± 2.67 ng/mL displayed as percentage of the baseline, *p* < 0.001, Figure 1B) compared to those without (group A, 2.29 ± 3.33 ng/mL vs. 2.33 ± 1.35 ng/mL displayed as a percentage of the baseline, *p* = 0.948, Figure 1B). Utilizing the Wilcoxon test and the Friedman test, highly significant results were given for the total population (*p* < 0.001, respectively) as well as the subgroup δ (*p* < 0.001, respectively). In contrast CD40L did not differ within the total population throughout the study period (253.46 ± 348.78 pg/mL vs. 301.91 ± 497.90 pg/mL displayed as a percentage of the baseline, *p* = 0.11, Figure 1A). However, when stratifying for a performance gain of ≥3% (group B = subgroups β and δ), differences between individuals with (255.07 ± 398.58 vs. 292.31 ± 420.72 pg/mL displayed as a percentage of the baseline, *p* = 0.001, Figure 1B) and without (group A, 249.27 ± 161.55 vs. 327.18 ± 668.88 pg/mL displayed as a percentage of the baseline, *p* = 0.472, Figure 1B) a described gain can be deduced. Especially subgroup β seems to drive these findings (323.10 ± 585.67 vs. 374.19 ± 615.97 pg/mL displayed as a percentage of the baseline, *p* = 0.001, Figure 2B) as subgroup δ displays a borderline significant *p*-value (199.26 ± 75.73 vs. 225.12 ± 79.13 pg/mL, *p* = 0.085, Figure 2B). The Wilcoxon test was found to be significant for the total cohort (*p* = 0.012) as well as subgroup β (*p* = 0.02). Pre-existing risk factors showed almost no correlation with RANTES and CD40L values in the respective subgroups. Only RANTES at the baseline and overweight in subgroup γ exhibited a tangible correlation (Spearman r = −0.484, *p* = 0.042, Appendix A Appendix A). Details regarding the progression of the abovementioned parameters, including their standard deviations, can be found in Appendix A Appendix A.

### 3.3. RANTES, CD40L, and the Development of Inflammatory Biomarkers

Subgroup α displayed a positive relationship between RANTES and calcitonin at the baseline (*p* = 0.036, r = +0.700, Appendix A Appendix A) and negative ones for IL6 (*p* = 0.013, r = −0.783, Appendix A Appendix A) and hsCRP (*p* = 0.008, r = −0.814, Appendix A Appendix A). None of these correlations prevailed at the end of the study. Furthermore, subgroup δ showed a positive association of RANTES and calcitonin at the end of the study (*p* = 0.014, r = +0.392, Appendix A Appendix A). A positive relation was noted for CD40L and calcitonin in subgroup δ (*p* = 0.015, r = +0.386, Appendix A Appendix A) and for hsCRP in subgroup γ (*p* = 0.019, r = +0.545, Appendix A Appendix A). Within the total cohort, no associations of RANTES or CD40L and the named inflammatory biomarkers were deducible (data not shown). However, when stratifying for performance gain of ≥3% (group B = subgroups β and δ) versus <3% (group A = subgroups α and γ), clear differences in the development of those markers appeared to become visible. Similarly to the changes of RANTES and CD40L, probands below the cutoff displayed no significant progression of calcitonin (2.40 ± 2.96 and 2.51 ± 2.60 pg/mL = displayed as a percentage of the baseline, *p* = 0.363), IL6 (2.56 ± 2.06 and 1.76 ± 1.23 pg/mL displayed as a percentage of the baseline, *p* = 0.072), and hsCRP (0.17 ± 0.25 and 0.13 ± 0.11 mg/dL displayed as a percentage of the baseline, *p* = 0.496) throughout the study (Figure 3A). In contrast, individuals with pronounced performance gain showed increased RANTES, CD40L, and calcitonin levels (2.23 ± 2.47 and 3.08 ± 3.06 pg/mL displayed as a percentage of the baseline, *p* < 0.001), as well as decreased IL6 levels (2.61 ± 1.63 and 2.11 ± 1.80 pg/mL displayed as a percentage of the baseline, *p* = 0.018) throughout the observational window (Figure 3B). An overview of changes regarding RANTES, CD40L, and the inflammatory markers within the individual subgroups can be found in Appendix A Appendix A–C. Details regarding the progression of the abovementioned parameters, including their standard deviations can be found in Appendix A Appendix A.

### 3.4. RANTES, CD40L, and the Development of Serum Lipids

Subgroups α (*p* = 0.050, r = −0.667, Appendix A Appendix A) and δ (*p* = 0.013, r = +0.393, Appendix A Appendix A) displayed correlations of LDL with RANTES at the baseline but not at the end of the study. Furthermore, CD40L was found to be associated with LDL in subgroup β at the baseline (*p* = 0.028, r = +0.389, Appendix A Appendix A) and in subgroup γ at the end of the study (*p* = 0.037, r = +0.494, Appendix A Appendix A). In group B (subgroups β and δ), there was a positive correlation of RANTES (*p* = 0.017, r = +0.283, data not shown) as well as CD40L (*p* = 0.003, r +0.346, data not shown) with LDL at the baseline but no significant correlation at the end of the study. Analyzing the progression of serum lipids throughout the study, similarities to the development of RANTES and CD40L seem to arise. Probands with a performance gain of <3% (Group A) showcased no changes in RANTES, CD40L, serum triglycerides (125.58 ± 78.25 and 142.51 ± 83.02 mg/dL displayed as a percentage of the baseline, *p* = 0.127), LDL (116.69 ± 37.25 and 117.53 ± 32.60 mg/dL displayed as a percentage of the baseline, *p* = 0.678), and HDL (58.58 ± 15.34 and 62.15 ± 16.56 mg/dL displayed as a percentage of the baseline, *p* = 0.054) levels with the observational window (Figure 4A). On the other hand, individuals with a performance gain of ≥3% (Group B) showed increased levels of RANTES, CD40L, and HDL (58.49 ± 18.29 and 61.37 ± 18.20 mg/dL displayed as a percentage of the baseline, *p* = 0.018), as well as decreased levels of LDL (116.61 ± 33.53 and 110.58 ± 31.96 mg/dL displayed as a percentage of the baseline, Figure 4B). An overview of changes regarding RANTES, CD40L, and the serum lipids within the individual subgroups can be found in Appendix A Appendix A–C. Details regarding the progression of the abovementioned parameters, including their standard deviations, can be found in Appendix A Appendix A.

## 4. Discussion

### 4.1. Exercise-Induced Changes of RANTES and Inflammatory Markers

Previous reports already demonstrated the short-term effects of physical exercise on circulating levels of RANTES. Through multiple measurements over 8 months, we were able to accurately picture the development under the conditions of long-term physical activity. RANTES displayed a significant, almost linear, increase within the total cohort. This effect was found to be even more pronounced in probands with a performance gain of ≥3% (group B) as they exhibited an average percentage increase of about 70% over the course of the study. Above all, initially athletic participants with a performance gain ≥ 3% (subgroup δ) showed a pronounced average gain of approximately 210%. These findings suggest an interdependency of RANTES and the performance level throughout the study, as well as basic fitness. The provided data are in conjunction with the work of Hoff et al., who described an increase under the conditions of bed rest (simulated weightlessness) and exercise for 60 days in 17 male probands [12]. Comparable baseline values are present in the mentioned publication (2–3 ng/mL) and our cohort (2.1 ng/mL). Baturcam et al. highlighted an increased RANTES expression in adipose tissue of obese compared to lean humans. These initially high levels decreased after 3 months of supervised physical exercise. Interestingly, RANTES levels in PBMCs were reported to be unaltered [11]. A prior publication of the research group stated there was no effect of physical exercise on circulating RANTES levels in obese humans [28]. Since low-grade inflammation is known to be present in obesity, the reported data may be put into perspective with our findings as well as the mentioned literature. Probands without a significant performance gain (group A) exhibited no changes in their inflammatory profile. On the other hand, in participants with a performance gain of ≥3% (group B), a pronounced decrease in IL6, mirroring the increase in RANTES, was deducible throughout the study. As hsCRP was found unchanged, calcitonin levels rose within the first months and reached a plateau afterward. These results seem plausible given the relatively low initial hsCRP levels of 0.19mg/dL on average and the fluctuation under physical exercise. Although calcitonin and procalcitonin are involved in various inflammatory reactions, elevated levels may be induced by another underlying cause. Calcium serum levels were found to be increased under the influence of physical exercise [29]. In group B, there was an observed elevation from 2.32 mmol/L to 2.36 mmol/L within the first 2 months of the study. Taking this into account, plateauing calcitonin levels could be the consequence of a metabolic change as opposed to a sign of heightened inflammation. Looking at the rise in RANTES levels and the decreased inflammatory profile (characterized by a significant reduction of circulating IL6), a two-sided picture appears for probands with a performance gain of ≥3% (group B). This threshold of ≥3% was initially set to select individuals with an adequate training success and all classically associated immunological and metabolic changes. As RANTES was reported to play a pivotal role in host defense, in particular within the adaptive immunity, enhanced serum levels could prove beneficial. Simultaneously reduced inflammatory markers suggest a specific, exercise-induced effect rather than a generalized pro-inflammatory setting. In perspective of the important link between physical activity and appropriate functions of the immune system, our data might help shed some light on this area. Although the nature of an observational study prevented us from deriving definite mechanistic conclusions, an increase in RANTES in a seemingly low inflammatory setting caused by physical exercise merits further studies.

### 4.2. Exercise-Induced Changes of CD40L

Although CD40L showed a delayed rise over the study period, no statistically significant changes were deducible within the total cohort. However, probands with a performance gain of ≥3% (group B) displayed markedly elevated CD40L serum levels at the end of the study. Further stratification showed that these findings were primarily driven by individuals of subgroup β (initially unathletic and performance gain ≥3%), whereas results in subgroup δ (initially athletic and performance gain ≥3%) were only borderline significant (*p* = 0.085). In addition to a weak correlation of RANTES at the baseline and overweight in subgroup γ, no correlations with pre-existing risk factors were deducible for RANTES or CD40L. There is evidence for a short-term decrease in CD40L under acute physical exercise [22]. Moreover, Bjørnstad et al. described a decrease in 15 participants suffering from heart failure (NYHA II–III) after a 20-week training regime [23]. These findings seem contradictory at first glance; however, an explanation may be found in the underlying pathology. Chronic heart failure (CHF) is known to be associated with an inflammatory status [30]. In the study of Bjørnstad et al., 20 healthy, age-, and gender-matched controls were included. These healthy probands displayed clearly lower baseline levels of CD40L (0–2500 pg/mL) compared to participants with CHF (above 10,000 pg/mL on average) [23]. In this regard, a decrease of the initially increased levels could hint toward a positive effect of physical exercise on the underlying inflammatory process. The baseline values in our total cohort were 253.48 pg/mL on average and therefore comparable to healthy controls but also apparently lower than those in probands with CHF. Since we did not include probands with heart failure or other known inflammatory conditions such as present infection or oncological diseases, these values seem to reflect the situation in an unselected cohort of individuals with at least one cardiovascular risk factor. As with RANTES, CD40L was reported to exert a key function in mounting an adequate immune response. Lack of CD40L/CD40L function results in a severely impaired B-cell response and opportunistic infections [18]. As previously discussed, enhanced circulating levels could benefit host defense, particularly in bolstering specific immune responses. In contrast to the linear increase in RANTES, CD40L displayed a slower, rather asymptotic progress. Individuals with high initial performance (>100%) tended to exhibit unaltered serum levels throughout the observational window. On the other hand, individuals with low initial performance (<100%) were prone to an increase over time. Individuals in group B (performance gain of ≥3%) showed significantly increased CD40L levels at the end of the study, whereas individuals in group A (performance gain of <3%) did not. Therefore, a performance-dependent component seems to be involved after all. RANTES displayed a linear rise in conjunction with the steady increasing performance throughout the study period. On the contrary, CD40L appears to be regulated through performance in a more binary way, as serum levels tended to stay unaltered or change in an asymptotic shape. As discussed previously, in group B (performance gain of ≥3%), improving performance was accompanied by an increase in CD40L as well as a decrease in IL6 serum levels. Taking the mentioned low baseline values into account, an exercise-induced effect seems more likely than an increased inflammatory status. Again, no final mechanistic conclusions can be drawn from this. However, a potential conjunction of physical activity and the adaptive immune system can be pointed out.

### 4.3. Influence of RANTES and CD40L on Serum Lipids

A connection of RANTES and CD40L with serum lipids was previously described in the literature [31,32,33]. In our cohort, RANTES showed correlations with serum LDL at the baseline in group B (*p* = 0.017, r = +0.283) as well as the subgroups α (*p* = 0.050, r = −0.667) and δ (*p* = 0.013, r = +0.393). Furthermore, CD40L displayed correlation with LDL in subgroup β at the baseline (*p* = 0.028, r = +0.389). One might deduce an involvement of both biomarkers in the metabolism of LDL at a steady state and before the initiation of the study-induced physical exercise. However, evidence for this finding appears weak and only observational. Analyzing the progression of serum lipids throughout the study, probands in group A (performance gain of <3%) showcased no changes in serum triglycerides, LDL, and HDL levels. Oppositely, individuals in group B (performance gain of ≥3%) exhibited significantly increased HDL and decreased LDL levels. These changes in circulating lipid levels are typically associated with the long-term effects of physical activity. Since almost no correlations were found with RANTES and CD40L during and at the end of the study, an interrelationship under the conditions of exercise was deemed as unconvincing. Although we were able to describe a loose connection of RANTES and CD40L with LDL at the baseline, no evidence for a deeper interconnection could be generated

## 5. Conclusions

Within the present study, we were able to picture an accurate progress of RANTES and CD40L serum levels throughout 8 months of physical activity. Previous studies already highlighted the short-term effects in smaller cohorts. However, to our knowledge, we provide the biggest available collective, with a total of 98 probands and the most extended observational window validated by two bicycle stress tests. The presented data comprise two major findings and one minor finding. First, under the conditions of continuous physical exercise, serum RANTES levels displayed a significant and linear increase within the total cohort. This finding was even more pronounced in individuals with a significant performance gain. Since those individuals exhibited a decreased inflammatory profile, a regulation through physical activity appears more likely than a generalized pro-inflammatory status. Implications on host defense seem probable but are beyond the scope of this study. However, enhanced circulating levels of RANTES and CD40L might be associated with a more potent response of the adaptive immune system. Especially factors such as viral clearance and increased resilience to tuberculosis as well as other infections could be the result of regular exercise. Second, throughout the study period, serum CD40L levels were found to be unaltered within the total cohort. However, probands with a significant performance gain showed an asymptotically shaped increase. Again, due to low baseline levels and decreasing IL6 concentrations, a modulation through physical training is deemed more likely than an underlying inflammatory cause. The observational nature of this study prevents definite mechanistic conclusions; nonetheless, modulation of the immune response seems presumable. Third, our data highlighted a loose association of RANTES and CD40L with LDL serum levels. These were found primarily at the baseline and did not prevail until the end of the study. Therefore, the influence of physical activity on serum lipid levels is supposed to be more likely than an interrelationship with RANTES or CD40L.

## Figures and Tables

**Figure 1 ijerph-19-08658-f001:**
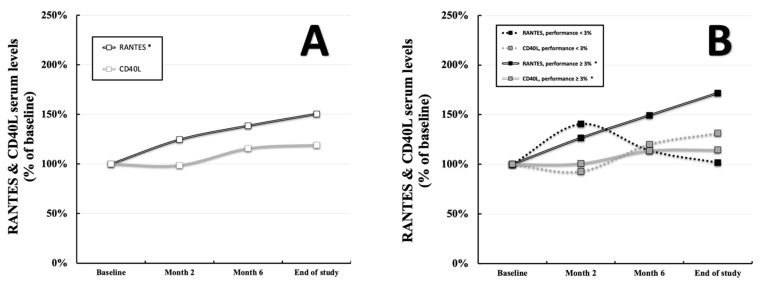
Progression of RANTES and CD40L serum levels within the total population (**A**) as well as the groups A (performance gain of <3%, *n* = 27, (**B**)) and B (performance gain of ≥3%, *n* = 71, (**B**)) over the course of 8 months. “*” indicates a two-tailed *p*-value of *p* < 0.05 comparing baseline values to those at the end of the study.

**Figure 2 ijerph-19-08658-f002:**
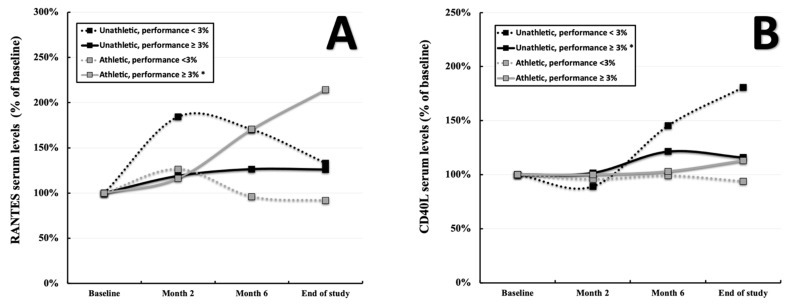
Progression of RANTES (**A**) and CD40L (**B**) serum levels within the subgroups α (initial performance < 100%, performance gain ≤ 2.9%, *n* = 9), β (initial performance < 100%, performance gain > 2.9%, *n* = 32), γ (initial performance ≥ 100%, performance gain ≤ 2.9%, *n* = 18), and δ (initial performance ≥ 100%, performance gain > 2.9%, *n* = 39) over the course of 8 months. “*” indicates a two-tailed *p*-value of *p* < 0.05 comparing the baseline values to those at the end of the study.

**Figure 3 ijerph-19-08658-f003:**
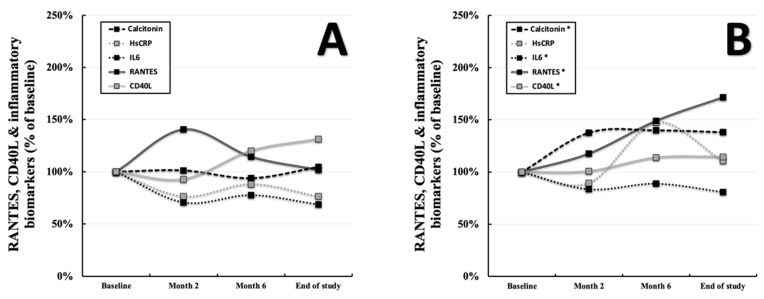
Progression of RANTES, CD40L, hsCRP, IL6, and calcitonin serum levels in groups A (performance gain of <3%, *n* = 27, (**A**)) and B (performance gain of ≥3%, *n* = 71, (**B**)) over the course of 8 months. “*” indicates a two-tailed *p*-value of *p* < 0.05 comparing baseline values to those at the end of the study.

**Figure 4 ijerph-19-08658-f004:**
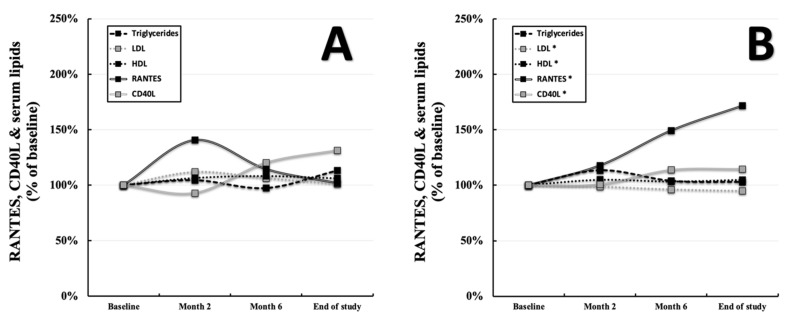
Progression of RANTES, CD40L, HDL, LDL, and serum triglycerides in groups A (performance gain of <3%, *n* = 27, (**A**)) and B (performance gain of ≥3%, *n* = 71, (**B**)) over the course of 8 months. “*” indicates a two-tailed *p*-value of *p* < 0.05 comparing baseline values to those at the end of the study.

**Table 1 ijerph-19-08658-t001:** Baseline characteristics of the total cohort as well as the subgroups α, β, γ, and δ. BMI = body mass index, HDL = high-density lipoprotein, LDL = low-density lipoprotein. Data are presented as median and interquartile ranges (25th–75th quartile) or percentages.

	Group α	Group β	Group γ	Group δ	Total Cohort
Parameters:	Gain ≤ 2.9% (*n* = 9)	Gain > 2.9% (*n* = 32)	Gain ≤ 2.9% (*n* = 18)	Gain > 2.9% (*n* = 39)	(*n* = 98)
Age (years)	50.3 ± 6.1	48.6 ± 7.9	50.4 ± 6.5	49.1 ± 6.0	49.3 ± 6.7
Female sex (%)	55.6%	46.9%	38.9%	28.2%	38.8%
Performance baseline (%)	87.4% ± 9.9%	88.8% ± 7.1%	122.0% ± 16.8%	116.0% ± 15.9%	105.6% ± 19.7%
Performance study end (%)	87.0% ± 9.1%	101.0% ± 10.0%	118.2% ± 18.0%	128.2% ± 15.6%	113.7% ± 20.0%
Performance gain (%)	−2.7% ± 4.3%	12.2% ± 7.1%	−3.8% ± 4.9%	12.1% ± 5.6%	7.8% ± 9.1%
Body composition:					
BMI (kg/m^2^)	27.8 ± 4.2	28.5 ± 5.2	27.2 ± 3.8	26.8 ± 3.3	27.5 ± 4.2
End of study BMI (kg/m^2^)	27.7 ± 4.6	28.2 ± 4.8	27.3 ± 4.1	26.7 ± 3.2	27.4 ± 4.1
Body fat (%)	33.9% ± 3.3%	31.6% ± 6.7%	26.8% ± 9.1%	27.8% ± 11.8%	29.4% ± 9.5%
End of study body fat (%)	31.5% ± 6.1%	29.7% ± 7.3%	26.7% ± 8.3%	23.4% ± 8.41%	26.3% ± 8.3%
Body muscle (%)	32.2% ± 3.7%	33.9% ± 4.1%	34.3% ± 3.8%	36.1% ± 4.0%	34.7% ± 4.1%
End of study body muscle (%)	32.4% ± 3.3%	34.3% ± 4.5%	34.4% ± 3.9%	36.2% ± 3.9%	34.9% ± 4.2%
Body water (%)	48.6% ± 2.4%	50.3% ± 4.9%	53.8% ± 6.7%	54.2% ± 5.9%	52.3% ± 5.9%
End of study body water (%)	50.4% ± 4.5%	52.1% ± 5.5%	53.9% ± 6.1%	56.3% ± 6.2%	53.9% ± 4.2%
Risk factors:					
Pack years	22.4 ± 21.4	18.9 ± 15.8	12.2 ± 9.2	16.3 ± 14.6	17.1 ± 14.9
Diabetes mellitus (%)	11.1%	3.1%	5.6%	0.0%	3.1%
Hypertension (%)	33.3%	43.8%	33.3%	23.1%	32.7%
Dyslipidemia (%)	33.3%	25.0%	38.9%	28.2%	29.6%
Overweight (%)	66.8%	68.8%	66.7%	63.2%	65.9%
Positive cardiac history (%)	11.1%	15.6%	5.6%	23.1%	16.3%
Positive family history (%)	66.8%	43.8%	50.0%	38.5%	44.9%
Laboratory values:					
Creatinine (mg/dL)	0.8 ± 0.1	0.8 ± 0.2	0.9 ± 0.2	0.9 ± 0.2	0.9 ± 0.2
Triglycerides (mg/dL)	154 ± 86	149 ± 100	111 ± 72	119 ± 62	131 ± 81
HDL-cholesterol (mg/dL)	52 ± 19	56 ± 22	62 ± 12	60 ± 15	59 ± 17
LDL-cholesterol (mg/dL)	126 ± 50	117 ± 32	112 ± 29	116 ± 35	117 ± 34
HbA1c (rel.%)	5.5% ± 0.4%	5.4% ± 0.8%	5.5% ± 0.9%	5.2% ± 0.3%	5.3% ± 0.6%
proBNP (pg/mL)	39 ± 27	59 ± 54	50 ± 35	32 ± 21	45 ± 39

## Data Availability

All data generated during this study are included in this manuscript and its Appendix A.

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
