# Peer review of "RANTES and CD40L under Conditions of Long-Term Physical Exercise: A Potential Link to Adaptive Immunity"

_ijerph, 2022, doi:10.3390/ijerph19148658_

Round 1

Reviewer 1 Report

This is an experimental study investigating the impact of chronic exercise training on inflammatory biomarkers, with a focus on RANTES and CD40L. The authors should be commended on conducting a training study in a very well sized cohort. Whilst the data collected provides a good basis for a good scientific manuscript, many elements within the manuscript require substantial attention.

Overall, the introduction is too long (it reads more like a generic literature review in places) and does not clearly outline the novelty of the current study. For example, the authors cite studies that looked at the effect of chronic exercise training on RANTES and CD40L – so what does the current study add?

In the methods, it is unclear whether participants actually performed the exercise that they were instructed to do. The authors say that they kept a training diary, but was this somehow monitored or evaluated? I’d also advise against naming groups “intervention” and “control”, maybe better “responder”/”non-responder”?

There is a very large amount of duplication in the results presented (e.g., including the same results into both text as well as figure format, or text and table format – some data are even presented twice in different figures (Fig 3 and 4). In general, data are over-analysed, which may lead to type II errors. For example, analysis is duplicated in many cases (analysis of raw data, analysis of percentage data – I would advise against analysing percentage converted data). Participants are grouped into multiple groups (A, B, alpha-gamma), again leading to analysis of the same data in different contexts. There is a vast amount of supplementary material, much of it looking at correlations. It is unclear whether the authors have somehow corrected for multiple analysis by way of post-hoc corrections (but as this is not reported I assume not). Unfortunately, the link to the supplementary materials did not work (www.mdpi.com/xxx/s1), so I cannot assess or comment on any of its contents.

Some standard measures are not reported, for examples standard deviations are not reported in any graphs. It is also not clear in the graphs what the * relates to exactly (e.g., Fig. 1A, does the P value related to pre vs end of study? In Fig. 1B, is it the difference between groups? – This unclarity applies to all figures).

Performance data is only given as a percentage of a calculated target value, it would be a lot more valuable to present measures in Watts (allowing comparison with existing literature). The figure labelling of the groups is not very user-friendly to the unfamiliar reader (alpha-gamma), figures would be a lot easier to grasp by the unfamiliar reader if the authors used a short descriptor (e.g., performance gain >3%, performance gain <3%, etc.)

Body composition measures in Table 1: only one value is reported per group (baseline), but was this not assessed over time (the authors make reference to this)? It would be worth including the post measures as well.

Language - whilst it is mostly clear what message the authors are trying to convey, I would recommend a native English speaker to check any future version of the manuscript.

Author Response

This is an experimental study investigating the impact of chronic exercise training on inflammatory biomarkers, with a focus on RANTES and CD40L. The authors should be commended on conducting a training study in a very well sized cohort. Whilst the data collected provides a good basis for a good scientific manuscript, many elements within the manuscript require substantial attention:

1) Overall, the introduction is too long (it reads more like a generic literature review in places) and does not clearly outline the novelty of the current study. For example, the authors cite studies that looked at the effect of chronic exercise training on RANTES and CD40L – so what does the current study add?

  • We thank the reviewer for the careful reading of the manuscript and the mentioned suggestions for improvement. We drastically shortened the introduction and removed the over-detailed sections on the background of RANTES and CD40L. Furthermore, the last paragraph was restructured to emphasize the novelty of the study:

“…Although there is the mentioned evidence on short-term effects of physical exercise in smaller collectives, more longitudinal data seems to be lacking. In contrast to the work of Hoff et al. as well as Bjørnstad et. al the present study provides a significantly longer observational period in a cohort of probands displaying a heterogeneous risk profile. [12, 23] we evaluated the changes of RANTES and CD40L in 98 participants undergoing eight months of physical exercise. To our knowledge we thereby provide the longest observational window for both biomarkers in the largest present cohort so far…”

  • Additionally it was outlined that the cohort of Bjørnstad et al. [23] consisted of patients suffering from chronic heart failure. The cohort of Baturcam et al. [11] was characterized as patients suffering from obesity. We therefore implemented a statement describing our cohort as a “cohort of probands displaying a heterogeneous risk profile“. According to the suggestions of the reviewer we believe that the executed changes significantly improved the introduction section and helped to underline the novelty of the study.

2) In the methods, it is unclear whether participants actually performed the exercise that they were instructed to do. The authors say that they kept a training diary, but was this somehow monitored or evaluated? I’d also advise against naming groups “intervention” and “control”, maybe better “responder”/”non-responder”?

  • We thank the reviewer for the comment. By restructuring the paragraph on bicycle stress tests we aimed to clarify the monitoring of physical exercise:

“…Probands were asked to keep a training diary, which was evaluated during each study visits to check the amount of conducted physical exercise. However, since the physical exercise was exerted without supervision, a thorough control of adherence was not possible. Therefore, a second bicycle stress test was performed at the end of the study to quantify changes in performance gain objectively…”

  • Additionally, we changed the classifications to “responder” and ”non-responder” as recommended by the reviewer.

3) There is a very large amount of duplication in the results presented (e.g., including the same results into both text as well as figure format, or text and table format – some data are even presented twice in different figures (Fig 3 and 4). In general, data are over-analysed, which may lead to type II errors. For example, analysis is duplicated in many cases (analysis of raw data, analysis of percentage data – I would advise against analysing percentage converted data). Participants are grouped into multiple groups (A, B, alpha-gamma), again leading to analysis of the same data in different contexts. There is a vast amount of supplementary material, much of it looking at correlations. It is unclear whether the authors have somehow corrected for multiple analysis by way of post-hoc corrections (but as this is not reported I assume not). Unfortunately, the link to the supplementary materials did not work (www.mdpi.com/xxx/s1), so I cannot assess or comment on any of its contents.

  • We thank the reviewer for the detailed assessment of the results section. Duplicate results were removed from the text section in reference to Table 1 and the respective figures. Moreover, percentages in the text were removed in order to avoid duplicate information. Since the figures represent data in percentages we included an explanation following the actual measured serum levels stating “…displayed as percentage of baseline in Figure..”. We agree with the reviewer on the possibility of type II error occurrence through extensive analysis. We performed analysis exclusively on raw data and emphasized this circumstance in the methods section “…Statistical analyses were exclusively done on non-standardized data, for the purpose of better representability, in figures with multiple parameters, data is presented as percentages of baseline values…”. To further reduce duplicate data and presentation of similar data in different contexts figures were revised. To keep clear comparability we removed duplicate figures in favor of presentation in percentages. All non-standardized data are available in the text of the results section, however figures now solely present percentage data. Post-hoc corrections were performed for ANOVA using Bonferroni correction, no post-hoc corrections were used looking at correlations. We are sorry for the inconvenience of the supplementary materials being not retrievable and will reupload them in the process of this revision. Overall we would like to thank the reviewer for helping us to improve clarity and structure of the results section.

4) Some standard measures are not reported, for examples standard deviations are not reported in any graphs. It is also not clear in the graphs what the * relates to exactly (e.g., Fig. 1A, does the P value related to pre vs end of study? In Fig. 1B, is it the difference between groups? – This unclarity applies to all figures).

  • We thank the reviewer for the comment. Due to the overlapping time points and the partially high standard deviations at mentioned overlaps, we decided to restrain from a visual representation of the standard deviations within the graphs. This was done for the purpose of clarity, however the newly added supplementary table 4 depicts the progression of all parameters including their standard deviations.

“…Details regarding the progression of the above-mentioned parameters, including their standard deviations can be found in Suppl. Table 4…”

  • In all figures * relates to changes from baseline to the end of the study. This fact was further emphasized and clarified within the respective figure legends.

5) Performance data is only given as a percentage of a calculated target value, it would be a lot more valuable to present measures in Watts (allowing comparison with existing literature). The figure labelling of the groups is not very user-friendly to the unfamiliar reader (alpha-gamma), figures would be a lot easier to grasp by the unfamiliar reader if the authors used a short descriptor (e.g., performance gain >3%, performance gain <3%, etc.).

  • We thank the reviewer for the suggestion of representing performance data as Watts. We agree on the benefits of comparability with existing literature and added the requested data to the manuscript (2.1.1. Baseline characteristics and performance gain). Additionally, figure labelling was changed to be more user friendly and easier to understand as suggested by the reviewer.

6) Body composition measures in Table 1: only one value is reported per group (baseline), but was this not assessed over time (the authors make reference to this)? It would be worth including the post measures as well.

  • We thank the reviewer for this comment and agree that longitudinal data on body composition at the end of the study period appears interesting. We therefore added the recommended data to Table 1.

7) Language - whilst it is mostly clear what message the authors are trying to convey, I would recommend a native English speaker to check any future version of the manuscript.

  • We thank the reviewer for the suggestion and consulted a native English speaker to improve grammar and readability of the final manuscript.

Reviewer 2 Report

Very nice work done by the authors. RANTES and CD40L were found increased by long-term physical exercise. I have provided specific comments to address on your paper, but most of them will be easy to address.  

My main concern was how this paper is considering the scope of this journal that the authors have not shown much appreciation. Moreover, the goal of the study needs to be properly highlighted and justified. Instead of setting their aim in the frame of a simple question, I would recommend that the authors attempt to present the key objectives of their study with regards to what is presently known (i.e. literature), thus highlighting the added value of the article.

Minor comments

The authors should definitely elaborate on the hypothesis as they are not sufficiently backed with theoretical considerations.

The program lasted 8 months. Why did you choose this period?

I recommend authors to describe the participants’ characteristics and how they were recruited?  

It would be appreciated if the authors could give more details about practical implications of the study

Author Response

Very nice work done by the authors. RANTES and CD40L were found increased by long-term physical exercise. I have provided specific comments to address on your paper, but most of them will be easy to address.  

1) My main concern was how this paper is considering the scope of this journal that the authors have not shown much appreciation. Moreover, the goal of the study needs to be properly highlighted and justified. Instead of setting their aim in the frame of a simple question, I would recommend that the authors attempt to present the key objectives of their study with regards to what is presently known (i.e. literature), thus highlighting the added value of the article.

  • We thank the reviewer for the comment and the suggestions. Regarding the scope of the journal, we aimed to prepare the manuscript for the special issue “Effects of Sport on the Immune System” as we hoped to appeal to the readers by providing insight into a potential link of physical exercise and biomarkers of adaptive immunity. We restructured the introduction section to reduce extensive background information in favor of a better structure. Especially within the last paragraph we aimed to emphasize to goal of the study and define its purpose in contrast the existing literature. Furthermore, the conclusion section was revised to give an outlook on potential results of increased RANTES and CD40L levels in term of host defense:

“…Since those individuals exhibited a decreased inflammatory profile, a regulation through physical activity appears more likely than a generalized pro-inflammatory status. Implications on host defense seem probable but are beyond the scope of this study. However, enhanced circulating levels of RANTES and CD40L might be associated with a more potent response of the adaptive immune system. Especially factors such as viral clearance and increased resilience to tuberculosis as well as other infections could be the result of regular exercise…”

  • According to the suggestions of the reviewer we believe that the structure of the manuscript was substantially improved and the key messages, the potential outlook, and the contrast to the existing literature was emphasized.

Minor comments

The authors should definitely elaborate on the hypothesis as they are not sufficiently backed with theoretical considerations.

  • We thank the reviewer for the answer and aimed to elucidate on the hypothesis in the newly revised introduction section. Especially the potential outlook of a link between regular physical exercise and enhanced functions of the adaptive immune system was implemented in the last paragraph:

“…To our knowledge we thereby provide the longest observational window for both bi-omarkers in the largest present cohort so far. Moreover, bicycle stress tests were performed at baseline and end of the study to set the actual performance levels in perspective to the circulating concentrations of both parameters. Potential findings of the present study could shed some light on the link of long-term physical exercise and the adaptive immune system. . Especially enhanced levels of RANTES and CD40L might hint towards a beneficial influence on exerted immune responses such as viral clearance and increased resilience to tuberculosis as well as other infections…”

The program lasted 8 months. Why did you choose this period?

  • We thank the reviewer for the question and are happy to answer it. Changes in body composition and exercise dependent markers such as plasma HDL and triglycerides were described as early as 5 months into regular physical exercise (1). Furthermore, resistance training was reported to reduce inflammatory cytokines 3 months into the training process (2). By choosing an observational period of 8 months, we aimed for a sufficient study duration in accordance with the previously described changes as well as other reports found in the literature. Moreover, due to limited financial support (indicated in the Funding section of the manuscript) a planned duration of more than 8 months was not realizable.

(1) Couillard C, Després JP, Lamarche B, Bergeron J, et al. Effects of endurance exercise training on plasma HDL cholesterol levels depend on levels of triglycerides: evidence from men of the Health, Risk Factors, Exercise Training and Genetics (HERITAGE) Family Study. Arterioscler Thromb Vasc Biol. 2001 Jul;21(7):1226-32

(2) Silva BSA, Lira FS, Rossi FE, et al. Inflammatory and Metabolic Responses to Different Resistance Training on Chronic Obstructive Pulmonary Disease: A Randomized Control Trial. Front Physiol. 2018 Mar 23;9:262.

I recommend authors to describe the participants’ characteristics and how they were recruited?  

  • We thank the reviewer for the suggestion and restructured the section “Characterization of the study cohort” (2. Materials and Methods). We emphasized the voluntary nature of participation and way of recruitment via pin-board notices. Additionally we stated that no monetary incentives were offered for the participation in the study.

“…A total of 109 probands took part in the presented study. Voluntary participants were recruited via pin-board notices, providing information of study design and inclusion criteria. No monetary incentives were offered for the participation in this study. Inclusion criteria were defined as the mental and physical capability to undergo bicycle stress tests and continuous endurance training for the duration of the study period…”

It would be appreciated if the authors could give more details about practical implications of the study

  • We thank the reviewer for the comment. We added a brief outlook on potential findings of the study into the revised introduction section. Furthermore we discussed the findings in terms of potential practical implications in the restructured conclusions section whilst still mentioning the limitations of an observational study preventing definite mechanistic insights.

Reviewer 3 Report

It is a well-structured article with a very clear question and where the authors manage to reach clear answers to the questions that are asked.

In the keywords, I suggest you not to use terms already used in the title. The use of repeated terms makes it lose strength in traditional search engines. I suggest you put other terms to give more strength to the searches

In the mentions to the authors, it would be advisable to put the year of publication

Author Response

It is a well-structured article with a very clear question and where the authors manage to reach clear answers to the questions that are asked.

1) In the keywords, I suggest you not to use terms already used in the title. The use of repeated terms makes it lose strength in traditional search engines. I suggest you put other terms to give more strength to the searches In the mentions to the authors, it would be advisable to put the year of publication

  • We thank the reviewer for this helpful suggestion and changed the keywords We agree on a potential loss in strength in regard to traditional search engines and restrained from duplicate words in the title and the keywords.

New keywords: Sport; training; CCL5; sCD40L; inflammation; immune system; 2022.